# The Effect of Leadership Styles and Initiative Behaviors of School Principals on Teacher Motivation

**Servet Yalçınkaya [1,\*]**, **Gökmen Dağlı [2,3]**, **Fahriye Altınay [4]**, **Zehra Altınay [5]** and **Ümit Kalkan [6]**

[1] Department of Educational Administration, Faculty of Education, Near East University, Nicosia 99010, Cyprus

[2] Institute of Educational Sciences, Department of Educational Science, Near East University, Nicosia 99010, Cyprus; gokmen.dagli@neu.edu.tr

[3] Department of Educational Science, University of Kyrenia, Nicosia 99010, Cyprus

[4] Institute of Graduate Studies, Societal Research and Development Center, Faculty of Education, Near East University, Nicosia 99010, Cyprus; fahriye.altinay@neu.edu.tr

[5] Societal Research and Development Center, Faculty of Education, Near East University, Nicosia 99010, Cy-prus; zehra.altinaygazi@neu.edu.tr

[6] Turkey's Ministry of National Education (MoNE), 06 648 Ankara, Turkey; umitkalkan@gmail.com

\* Correspondence: servetyalcinkaya@gmail.com; Tel.: +90-533-844-33-01

**Abstract:** This study aims to investigate the effect of school administrators' personal initiative behaviors and their leadership styles on teacher motivation. In this study, designed with a quantitative research approach, the relational scanning model was used, and a model was created to test the effect of school administrators' leadership styles and personal initiative behaviors on teacher motivation. In this context, the leadership styles of school administrators and their personal initiative behaviors were studied as independent variables, while teacher motivation was studied as a dependent variable. In the study, 406 teachers working in high schools in Manisa city center were selected using the purposeful sampling method. Pearson moment correlation techniques and multiple regression analysis techniques were used to determine whether there was a significant relationship between dependent and independent variables during the data analysis. According to the results obtained from the research findings, it was observed that the general motivation levels of the teachers were high. In the analyses conducted in terms of the relationships between school administrators' personal initiative-taking behaviors and teachers' motivation, it was found that these variables had a significant and positive relationship; similarly, it was determined that there is a significant relationship between the leadership styles of school administrators and the motivation of teachers. In the regression model established to examine the effect of the leadership styles of school administrators on the internal factor dimensions of teacher motivation, it was determined that all sub-dimensions of leadership styles predicted teachers' internal motivation. This research provides evidence that school administrators' personal initiative behavior is directly related to teacher motivation and positively affects teacher motivation, thereby affecting the quality of their teaching.

**Keywords:** motivation; teacher motivation; personal initiative; leadership styles; school principal

## 1. Introduction

Today, business conditions are increasingly competitive and the focus on success makes employee motivation important both individually and organizationally. While motivation affects the morale of employees, their attitudes and behaviors towards the organization, it is also an important factor in achieving individual and organizational goals. Directing individuals with different characteristics to unite around a common goal and achieve success within an organization is only possible if the employees are properly coordinated and motivated by the manager [1]. From this point of view, management practices in schools, which include individuals with different characteristics and are one of

the largest organizations, can be considered among the main factors that positively affect teachers' motivation.

Schools are at the forefront of organizational structures whose products are "human". Schools, which undertake the collective education function of individuals with all kinds of education and training activities [2], are among the organizational structures that have the most communication with a society and are comprised of various components, including students, parents and the school environment. One of the basic elements of this important organizational structure is, undoubtedly, school principals.

School administrators, who are the main actors in a school administration, are a team consisting of the principal, chief assistant principal and assistant principals in Turkey. However, the person who is seen as the leader in the school is primarily the school principal. The principal of the school, who takes duty by appointment, is the administrator who receives power from legal, social and technical powers. The school principal takes their legal authority from the authority they hold, social powers from the staff and environment they work in, and their technical powers from their knowledge and management skills [3]. The school principal is expected to have all of these powers to be an effective leader. Furthermore, they must have some other characteristics in order to perform their duties successfully. One of the most prominent of these is the leadership skill. This is because the school administration should be able to use its leadership qualities effectively to motivate teachers, unite them around organizational goals and, most importantly, to improve education training [4,5].

### 1.1. Leadership Styles

The concept of leadership has been defined in different ways by researchers, and has been analyzed using various theories and approaches. Of course, these concepts have differed according to the characteristics of the current century and the social, political and cultural characteristics of society at various points in time [6,7]. The leadership styles classification developed by Lewin, Lippit, and White (1939), which is one of the behavioral theories, was used as a source for this research. This classification has been explained in the literature as democratic, autocratic, and full-freedom leadership styles.

In the democratic leadership style, the organization manager includes all stakeholders in the management and decision-making mechanism. In regard to democratic leadership, which is a participatory leadership type, researchers state that this is one of the most effective methods. Group members have high productivity and leaders are effective in the motivation of employees [8]. Democratic leaders always communicate with group members and find their views valuable in determining the organization's goals, plans and policies, and in dividing up work [9–12]. In the autocratic leadership style, managers derive their "management power" from laws. All decisions in the organization are taken by the manager and subordinates are expected to implement the decisions. Managers do not consult with stakeholders in decisions regarding the organization, and simply want them to follow orders without waiting for an explanation [8]. Autocratic leaders closely monitor employees' performance; they encourage competition among employees, reward success and punish poor performance. Autocratic leaders are often directive and task-oriented [13]. In a leadership style that allows full freedom, the employees of the organization are free to work as they please and there is no intervention in decision making and intra-organizational conflicts. This is the least preferred leadership style by managers and is based on an "avoidant" approach [14]. There is no hierarchy of authority in the organization. Leaders leave group members independent in many aspects, encourage them, and do not worry about the needs or well-being of employees [15]. This is a leadership style that allows employees to allocate resources to achieve their goals, plans and programs independently [11,16,17].

As a result, because the leadership styles of organizational managers vary, each of them may have different effects on employees. When the subject is considered in terms of educational management, it is known that the leadership style of a school principal is

one of the factors affecting the interest, attitudes and behaviors of the school, especially the sense of belonging and motivation instilled in the stakeholders (teachers) towards the school. The studies conducted also support this view. Kılıç [18], in his study examining the effect of the leadership styles of school administrators on teacher motivation, reported that school principals who adopted a democratic management style positively affected teachers' motivation.

*1.2. Personal Initiative*

Personal initiative behavior consists of three dimensions that complement each other: "self-initiation", "proactivity" and "persistence". Self-initiation refers to the occurrence of behavior without outside pressures, role requirements, instruction, or an overt action [19]. In this respect, the personal initiative requires the employee to establish his/her own goals rather than the goals determined by the organization. Proactivity means the employee anticipating problems and opportunities and taking action [20,21]. The proactive employee strives to change the business environment according to the needs of the future. Persistence also refers to the active struggle of the employee against the obstacles that may arise while implementing the goals set with a proactive approach [22]. A persistent manager tries to achieve his/her goal with new methods, regardless of the time spent and the number of attempts.

School administrators in Turkey carry out all activities related to their tasks according to the existing legislative provisions of the Ministry of National Education. Therefore, all school administrators are equally responsible for the work and transactions related to their field of duty and are subject to the same legislation. In the "Education Vision for 2023" document of the Ministry of National Education, basic expectations from school administrators about creating a corporate identity, capturing team spirit, being productive and managing the unforeseen are included. Furthermore, the success of all kinds of reforms and improvement efforts in areas such as the curriculum, materials and technology, especially in education policies, is largely attributed to the professional competencies, perceptions and commitment of teachers and school administrators.

The fact that some school administrators are more successful in schools may be due not only to them following the standard procedures, but also to the way in which they demonstrate their personal competencies and characteristics by exhibiting different behaviors beyond this. In this context, the contributions that administrators provide by going beyond the requirements of their roles in order to achieve the main goals of the school are considered as part of personal initiative. Frese [23] defines initiative as an active performance concept, expressing personal initiative as "interpreting the role of the employee, creating new goals for the benefit of the organization and applying these goals persistently". Fayol [24] states that a manager who takes initiative can increase the desire and energy of employees at all levels of the organization. Furthermore, the personal initiative should not be considered as an "overrun", but rather as revealing more than what is expected from the manager himself/herself [25]. In other words, personal initiative is the behavior of taking responsibility and taking action in line with the aims of the organization, without waiting for instructions from someone else.

Studies on the subject reveal that personal initiative has a positive effect on both organizations and individuals. Individuals with personal initiative achieve better academic results [26], become more entrepreneurial [27], find jobs more easily, and are more insistent about realizing their dreams. On the other hand, business environments that encourage personal initiative are seen to be more effective, profitable and change oriented [28]. However, personal initiative is not a subject that has been adequately addressed in the field of school administrators and organizational psychology, although managers in various fields have consistently emphasized the need for highly resourceful employees.

*1.3. Teacher Motivation*

Motivation can be expressed as the fulfilment of these conditions by researching the necessary conditions for the employees to work willingly in line with organizational goals and to be productive. In management research, motivation is generally defined as the movement to initiate, direct and maintain desired business behaviors. The main purpose of motivation is to ensure that employees act willingly and efficiently in parallel with the goals of the organization [29]. Therefore, high employee motivation directly affects organizational performance [30] and leads to organizational success.

In terms of educational institutions, the motivation of employees is slightly different from that of other organization employees, because there are three important manpower resources in the education system. Each of these resources, consisting of administrators, teachers and students, is in constant interaction with each other. While students have not yet acquired the desired qualifications, but are the most basic resources in the education system, teachers are a valuable element that processes these resources [31]. Education managers, on the other hand, are considered as the authorities empowered to use these resources more effectively for organizational purposes.

Teaching is an extremely important profession in terms of human sensitivities. Teachers know that they will shape individuals' futures, including their choice of profession, and they consider this sensitively [32]. On the other hand, teachers are the most basic human resources that can affect generations and society as a whole. Teachers' duty to provide learners with knowledge, skills and experiences makes teacher motivation a critical and important issue [33]. Therefore, teacher motivation is a very important driving force in the success and output of educational services. For teachers to overcome their specific duties and responsibilities, it is also necessary to meet their physiological, psychological and social needs so that they are ready to provide educational services. Furthermore, school administrators meeting the needs of teachers leads to a positive contribution in the form of an increase in employee motivation [34], encouragement for teachers to assist in the solving of institutional problems, and a more efficient working environment.

Studies examining this concept and, accordingly, its impact on teacher performance, having an important role in the effectiveness of schools, have noted that job satisfaction and the motivation of all staff, especially teachers, are positively affected in institutions where a positive school climate and social support are provided [35,36]. The aforementioned studies show that teacher motivation is influenced by internal factors stemming from teachers' working environments rather than external factors such as salary, education policy and education reforms. Although the national and international literature investigating the effects of some characteristics of school administrators on teachers' motivation is limited [20,37], the literature does make clear that the personal initiative behavior of administrators has a positive relationship with individual performance and organizational effectiveness.

As a result, it would be fair to state that improving the performance of school principals and increasing the effectiveness of schools are among the main problems of the educational management discipline. For this reason, it is seen as a meaningful effort to examine the administrative behaviors and competencies of school principals from the perspective of leadership styles and personal initiative. Moreover, ensuring the motivation of teachers and strengthening their commitment to the school would primarily be beneficial for the school. The results of this study, which was conducted to determine the motivation levels of teachers in the context of leadership styles and initiative behaviors of administrators, are important both for their contribution to the field of educational management and supervision, the future of organizations and the quality of education.

The aim of this study is to investigate the effects of school administrators' personal initiative behaviors and the leadership styles they display on teacher motivation. For this purpose, answers to the following questions were sought to determine the relationships between variables:

1.  What are the perceptions of school administrators' leadership styles, levels of taking personal initiative and teacher motivation?

2. Is there a significant relationship between school administrators' personal initiative behaviors and teacher motivation?
3. Is there a significant relationship between the leadership styles of school administrators and teacher motivation?
4. Do school administrators' personal initiative behaviors predict teacher motivation?
5. Do the leadership styles exhibited by school administrators predict teacher motivation?

## 2. Materials and Methods

Designed in the context of the quantitative research approach, this study is based on the relational screening model. Survey models are research approaches that aim to describe a past or present situation as it happens [38]. The relational survey model, on the other hand, is a research model that aims to determine the existence or degree of change between two or more variables [39], and it aims to describe the views and qualities of the masses.

In this study, a model was created to test the effect of school administrators' leadership styles and personal initiative behaviors on teacher motivation. In this context, the leadership styles of school administrators and their personal initiative were studied as independent variables, with the sub-dimensions of teacher motivation as dependent variables. The dependent and independent variables of the research are presented in Figure 1.

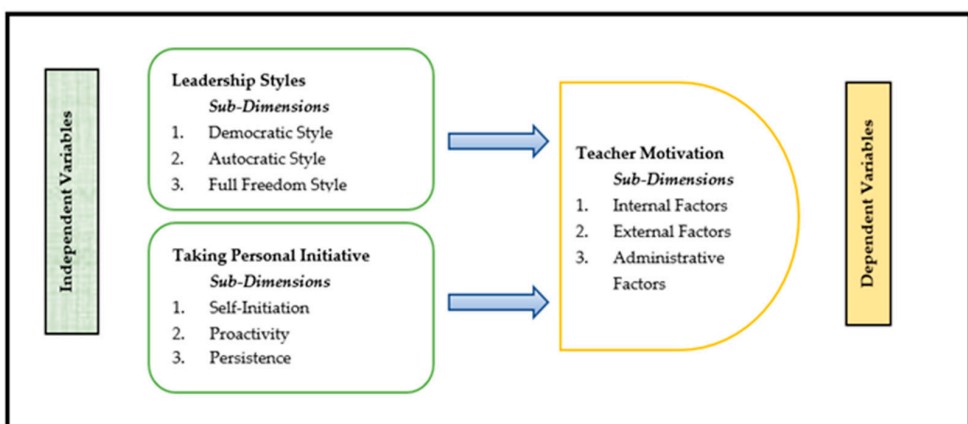

**Figure 1.** The dependent and independent variables of the research.

### 2.1. Population and Sample

The general population of this research is 6553 teachers working in high schools in Manisa city center in the 2019–2020 academic year. Due to the outbreak of COVID-19 experienced across the world, including Turkey, travel restrictions, time limitations and the risk of transmission, two major central districts of Manisa were selected as suitable sample locations. Therefore, 540 teachers working in 14 high schools in the Yunus Emre and Şehzadeler districts of Manisa province constitute the study sample. According to Fraenkel and Wallen [40], there are no exact values to determine a sample that represents the population. This depends on the research budget, energy and effort of the researcher. The "purposeful sampling" method was used while determining the sample of the study. Purposeful sampling is a non-random sampling type, and it is a sampling method that allows in-depth research by selecting information-rich situations, depending on the purpose of the study [38]. The current number of teachers and managers (principal, chief assistant principal and assistant principal) of the high schools within the scope of the study are presented in Table 1.

The current number of teachers at the high schools in the research sample is 540, and the number of administrators (principal, chief assistant principal and assistant principal) is 53. Since it is not possible to reach all of the participants, it was calculated that the number of participants should be at least 370 by "calculating the number of students to

be reached with sample size sampling error" [41], for a 95% confidence level and $\alpha = 0.05$ using the sample formula with a known population. The number of participants in this research is 406. Therefore, it was concluded that the sample size in the study was sufficient. Information on the demographic characteristics of the teachers is presented in Table 2.

**Table 1.** Current Number of Teachers and Managers of High Schools Belonging to the Research Sample.

| High Schools | Teacher | Manager |
|---|---|---|
| Manisa Science High School | 32 | 5 |
| TOBB Bülent Koşmaz Science High School | 15 | 2 |
| Gediz Anatolian High School | 28 | 3 |
| Fatih Anatolian High School | 35 | 3 |
| Social Sciences High School | 32 | 4 |
| Ş. F. K. Girls Anatolian Imam Hatip High School | 23 | 3 |
| Şehzadeler Girls Anatolian Imam Hatip High School | 42 | 6 |
| Yunus Emre Anatolian Imam Hatip High School | 30 | 5 |
| Manisa High School | 78 | 4 |
| Dündar Çiloğlu Anatolian High School | 48 | 3 |
| Halit Görgülü Anatolian High School | 36 | 3 |
| M. Efendi Vocational Technical Anatolian High School | 33 | 6 |
| Cumhuriyet Anatolian High School | 55 | 3 |
| Mehmet Akif Ersoy Anatolian High School | 53 | 3 |
| Total | 540 | 53 |

Source: T.C. MEB, Statistics Yearbook, 2019. http://sgb.meb.gov.tr/meb_iys_dosyalar/201909/30102730meb_istatistikleri_orgun_egitim_2018_2019 (accessed on 1 August 2020).

**Table 2.** Demographic Characteristics of Participants (N = 406).

| Variables | | f | % |
|---|---|---|---|
| Gender | Female | 220 | 54.2 |
| | Male | 186 | 45.8 |
| Age | 35 and less | 51 | 12.6 |
| | 36–40 age | 38 | 9.4 |
| | 41–45 age | 84 | 20.7 |
| | 46–50 age | 122 | 30 |
| | 51 and above | 111 | 27.3 |
| Seniority | 1–5 years | 44 | 10.8 |
| | 6–10 years | 56 | 13.8 |
| | 11–15 years | 73 | 18 |
| | 16–20 years | 109 | 26.8 |
| | 21 years and above | 124 | 30.5 |
| Education level | Undergraduate | 317 | 78.1 |
| | Postgraduate | 89 | 21.9 |
| Time spent working with the same principal | 1–5 years | 267 | 65.8 |
| | 6–10 years | 47 | 11.6 |
| | 11–15 years | 25 | 6.2 |
| | 16–20 years | 34 | 8.4 |
| | 20 years and above | 33 | 8.1 |
| Time spent working at the same school | 1–5 years | 212 | 52.2 |
| | 6–10 years | 111 | 27.3 |
| | 11–15 years | 37 | 9.1 |
| | 16–20 years | 21 | 5.2 |
| | 20 years and above | 25 | 6.2 |
| Branch category | Verbal Branches | 217 | 53.4 |
| | Quantitative Branches | 124 | 30.5 |
| | Vocational and Technical Branches | 31 | 7.6 |
| | Special Talent Branches | 34 | 8.4 |
| School type | Anatolian High School | 218 | 53.7 |
| | Vocational and Technical High School | 60 | 14.8 |
| | Imam Hatip High School | 56 | 13.8 |
| | Science High School | 46 | 11.3 |
| | Social Science High School | 26 | 6.4 |

## 2.2. Data Collection Tools

"Leadership Style Scale of School Administrators", "Taking Personal Initiative Scale of School Administrators" and "Teacher Motivation Scale" were used as data collection tools.

## 2.3. Leadership Style Scale of School Administrators

The leadership styles scale was developed by Kılıç and Yılmaz [42] to determine the perception levels of leadership styles exhibited by school administrators towards their employees. The scale consists of 16 items and three sub-dimensions: "democratic style" ($n$ = 9 items), "autocratic style" ($n$ = 4 items) and "style that allows full freedom" ($n$ = 3 items). The scale rated in five-point Likert form. It was scored as "totally agree = 5", "agree = 4", "partially agree = 3", "disagree = 2" and "totally disagree = 1". The Cronbach's alpha values of the scale [42] were calculated as 0.89 in the democratic style dimension, 0.82 in the autocratic style dimension, and 0.71 in the style dimension that allows full freedom. The reliability coefficients obtained for this research were found to be 0.97 in the democratic style dimension, 0.87 in the autocratic style dimension, and 0.81 in the style dimension that allows full freedom. The Cronbach's alpha value of the whole scale was found to be 0.91.

"Democratic style", which is the first sub-dimension of the leadership style scale, consists of statements that include democratic leadership behaviors such as school administrators being democratic and participatory, enjoying guiding teachers, encouraging them with regard to new projects, adopting team spirit, and being open to change in regard to developments. The second sub-dimension of the scale, the "autocratic style", relates to leadership behaviors such as school administrators making decisions alone, closely and tightly supervising employees/projects, making final decisions using their authority, and not giving teachers opportunities in any subject. The third sub-dimension of the scale, "style that allows full freedom", emphasizes leadership behaviors such as having a more liberal attitude towards employees, not supporting staff development or the adoption of new ideas, not worrying about the completion of work on time and not expressing an opinion on most issues.

## 2.4. Taking Personal Initiative Scale of School Administrators

The scale developed by Akın [25] to determine the initial status of school administrators has 32 items in total, and the scale consists of three sub-dimensions: "self-initiation" ($n$ = 13 items), "proactivity" ($n$ = 9 items) and "persistence" ($n$ = 10 items). Responses are rated on a five-point Likert scale: "totally agree = 5", "strongly agree = 4", "moderately agree = 3", "slightly agree = 2" and "strongly disagree = 1". The Cronbach's alpha values of the personal initiative scale [25] were calculated as 0.93 in the self-initiation dimension, 0.92 in the proactivity dimension and 0.94 in the persistence dimension. The reliability coefficients obtained for this study were found to be 0.95 in the self-initiation dimension, 0.95 in the proactivity dimension and 0.96 in the persistence dimension. The Cronbach's alpha value of the whole scale was found to be 0.95.

The first sub-dimension of the scale, "self-initiation", includes teachers' expressions about situations such as administrators ignoring legislation for the benefit of the school, taking important and critical decisions on their own, assuming important responsibilities, easily implementing their management ideas, offering suggestions to higher authorities for the better management of the school, encouraging employees to make new projects and applications, seeking better ways in which to perform their role (instead of standard practices) and wanting to see the changes he/she likes that have been implemented in other institutions also implemented in his/her own institution. The second sub-dimension of "proactivity" includes expressions such as school administrators constantly seeking to do their job better, looking for opportunities that will benefit the school, addressing problems, efforts to improve themselves in a professional capacity, turning problems into opportunities, and being excited to make changes. "Persistence", the third sub-dimension of the scale, includes expressions such as trying to accomplish the duties assigned to them by higher authorities, standing behind their work, struggling with the status quo when

it comes to innovation, not giving up in the face of obstacles, being patient, seeking new solutions in case of failure and enjoying overcoming obstacles.

### 2.5. Teacher Motivation Scale

The Teacher Motivation Scale was developed by Kılıç and Yılmaz [43] to determine the motivation levels of teachers towards the profession. The scale consists of 18 items and three sub-dimensions: "internal motivation" ($n$ = 8 items), "external motivation" ($n$ = 5 items) and "administrative motivation" ($n$ = 3 items). The scale rated in five-point Likert form; was scored as "totally agree = 5", "agree = 4", "partially agree = 3", "disagree = 2" and "totally disagree = 1". The Cronbach's alpha values of the scale [43] were calculated as 0.81 in the internal motivation dimension, 0.72 in the external motivation dimension, and 0.70 in the style that allows full freedom. The reliability coefficients obtained for this study were found to be 0.83 in the internal motivation dimension, 0.74 in the external motivation dimension and 0.85 in the style dimension that allows full freedom. The Cronbach's alpha value of the whole scale was found to be 0.85.

The first sub-dimension of the scale, "internal factors", contains teachers' perceptions of the financial and moral gains that this profession provides them and their adoption within the schools where they work, developing their talents and skills, their feelings and thoughts about the teaching profession, whether they have the authority to make their own decisions in regard to their profession, their ability to take the initiative, to what extent they feel that they work in a free environment, and to what extent they feel they are successful in their profession. "External factors", the second sub-dimension of the scale, covers the situations that develop outside individuals and motivate them towards their work. This sub-dimension tries to reveal the level of perception teachers have with regard to the respect afforded to them by society as a member of the teaching profession, retirement at the end of their working life, having appropriate physical equipment provided by the institutions they work for, working with experts in their field, and being provided with support in regard to health and social security. "Administrative factors", the third sub-dimension of the scale, includes the leadership attitudes and behaviors of school administrators towards teachers. This sub-dimension tries to determine teachers' opinions about various aspects, such as managers being more sensitive and helpful, teachers appreciating the work they do together, as well as their support, whether administrators encourage them to undertake new studies, the creation of a peaceful working environment and the trust between teachers and administrators.

### 2.6. Data Collection Process

The research data were collected in the spring semester of the 2019–2020 academic year. The necessary permissions were obtained from Manisa Provincial Directorate of National Education for the research. The scales were collected by the researchers, after providing the necessary explanations to the teachers working in the institutions participating in the study. Approximately 550 scales were distributed to schools. The collected scales were examined and 406 scales were evaluated, after those who were excluded from the evaluation were removed due to reasons such as having not filled out the survey correctly or completely, e.g., as a result of marking more than one option. Furthermore, the data collection process was carried out voluntarily by obtaining consent forms from the participants. Filling in their responses to the scales took approximately 20 min.

### 2.7. Analysis of Data

The SPSS for Windows 22 (Statistical packages for Social Sciences 22) statistical package program was used in the analysis of the research data. Firstly, 406 scales were transferred to the computer environment and a data set was created. Frequency and percentage values were calculated to determine the demographic characteristics of teachers (gender, age, education level, seniority, branch, time spent working with the same principal, and time spent working at the current school, and type of high school they work in). Before

starting the analysis of the research data, lost data analysis, linearity and normal distribution tests were applied. It was determined that the research variables did not show a normal distribution. The Pearson moment product correlation technique and multiple regression analysis were used to determine whether there was a significant relationship between the dependent and independent variables.

In our research, while interpreting the arithmetic averages regarding the leadership styles of school administrators, their personal initiative-taking behaviors and the motivation levels of the teachers, the ranges were found to be 1.00–1.79 "quite low", 1.80–2.59 "low", 2.60–3.39 "medium", 3.40–4.19 "high", while the range of 4.20–5.00 was evaluated as "quite high" [44].

## 3. Results

In this section, the findings obtained from the opinions of the teachers are given in line with the sub-dimensions of the research.

### 3.1. Findings Regarding Teachers' Views Relating to Research Variables

As seen in Table 3, teachers' perceptions of their administrators' leadership styles are "medium" ($\bar{x}$ = 3.394; ±0.730). It is seen that the perception of democratic leadership sub-dimension ($\bar{x}$ = 4.065; ±1.058) and the autocratic leadership sub-dimension ($\bar{x}$ = 3.708; ±0.989), which are sub-dimensions of the leadership styles scale, are at the "high" level, and the leadership perception that relates to full freedom is "low" ($\bar{x}$ = 2.319; ±1.223) level.

**Table 3.** Findings Regarding Leadership Styles, Initiative-Taking Behaviors of Administrators and Teacher Motivation According to Teacher Opinions.

| Variables | $\bar{x}$ | Ss |
|---|---|---|
| Leadership Styles | 3.394 | 0.730 |
| Democratic Leadership | 4.065 | 1.058 |
| Autocratic Leadership | 3.708 | 0.989 |
| Leadership with Full Freedom | 2.319 | 1.223 |
| Taking Personal Initiative | 4.009 | 0.865 |
| Self-Initiation | 3.961 | 0.877 |
| Proactivity | 3.985 | 0.907 |
| Persistence | 4.042 | 0.886 |
| Teacher Motivation | 4.816 | 0.645 |
| Internal Factors | 4.106 | 0.718 |
| External Factors | 4.200 | 0.715 |
| Administrative Factors | 4.282 | 0.764 |

Teachers' perceptions of school administrators' behaviors in regard to taking personal initiative ($\bar{x}$ = 4.009; ±0.865) are at a "high" level. Among the personal initiative behavior sub-dimensions, the self-initiation sub-dimension is $\bar{x}$ = 3.961; ±0.877, the proactivity sub-dimension is $\bar{x}$ = 3.985; ±0.907 and the persistence sub-dimension perception is $\bar{x}$ = 4.042; ±0.886. As such, teachers' perceptions about all sub-dimensions of the scale are at a "high" level.

When the items related to teachers' motivation are examined, it is seen that the total motivation scores ($\bar{x}$ = 4.181; ±0.645) are "high". When the sub-dimensions of the scale are examined, the internal factors sub-dimension ($\bar{x}$ = 4.106; ±0.718) is "high", while external factor ($\bar{x}$ = 4.200; ±0.715) and administrative factor sub-dimensions ($\bar{x}$ = 4.282; ±0.764) seem to be at a "quite high" level.

### 3.2. Findings on Relationships between Variables

In this section, our findings regarding the relationship between research variables and sub-dimensions are included. Firstly, the relationship between the sub-dimensions of school administrators taking personal initiative and the sub-dimensions of teacher motivation was examined. Afterwards, in line with teachers 'opinions, we examined whether there is a significant relationship between the sub-dimensions of school administrators' leadership styles and the sub-dimensions of teacher motivation.

The results of the correlation analysis between the sub-dimensions of teacher motivation and the sub-dimensions of school administrators taking personal initiative are presented in Table 4.

**Table 4.** Correlation Analysis Results between School Administrators' Personal Initiative-Taking Behaviors and Teacher Motivation.

| Sub-Dimensions | | Internal Factors | External Factors | Administrative Factors |
|---|---|---|---|---|
| Self-Initiation | $r$ | 0.533 ** | 0.547 ** | 0.674 ** |
| | $p$ | 0.000 | 0.000 | 0.000 |
| Proactivity | $r$ | 0.537 ** | 0.531 ** | 0.686 ** |
| | $p$ | 0.000 | 0.000 | 0.000 |
| Persistence | $r$ | 0.583 ** | 0.539 ** | 0.662 ** |
| | $p$ | 0.000 | 0.000 | 0.000 |

** $p < 0.001$.

As can be seen in Table 4, as a result of the correlation analysis carried out for the relationships between variables, according to teachers' opinions, there is a positive and significant relationship between the self-initiation, proactivity and persistence dimensions of the school administrators' taking initiative scale and the internal, external and administrative factor dimensions of the teacher motivation scale at the $p < 0.001$ level. The results of the correlation analysis between the leadership styles of school administrators and the scores of teacher motivation sub-dimensions are presented in Table 5.

**Table 5.** Correlation Analysis Results between Leadership Styles of School Administrators and Teacher Motivation.

| Sub-Dimensions | | Internal Factors | External Factors | Administrative Factors |
|---|---|---|---|---|
| Democratic Style | $r$ | 0.538 ** | 0.486 ** | 0.813 ** |
| | $p$ | 0.000 | 0.000 | 0.000 |
| Autocratic Style | $r$ | 0.517 ** | 0.506 ** | 0.588 ** |
| | $p$ | 0.000 | 0.000 | 0.000 |
| Full Freedom style | $r$ | 0.065 | −0.027 | −0.142 ** |
| | $p$ | 0.192 | 0.591 | 0.004 |

** $p < 0.001$.

As can be seen in Table 5, there is a positive and significant relationship between the democratic and autocratic style sub-dimensions of the leadership styles scale of school administrators and the internal, external and administrative factors of the teacher motivation scale, at the level of $p < 0.001$, according to teachers' views. On the other hand, it was determined that there was no significant relationship between internal and external factors of teacher motivation in the full freedom style sub-dimension, but a negative correlation at the level of $p < 0.004$ in the dimension of administrative factors.

### 3.3. Findings Related to Regression Analysis

In this section, according to teachers 'opinions, we examined whether school administrators' personal initiative-taking behaviors and leadership style sub-dimensions significantly predicted teacher motivation. For this purpose, a regression model was created to examine the effect of the personal initiative sub-dimensions of school administrators on internal, external and administrative factors of teacher motivation. The analysis results of the model are presented in Table 6.

As seen in Table 6, the regression model established to test the effect of the personal initiative behavior dimensions (self-initiation, proactivity, persistence) of school administrators on the dimension of internal factors of teacher motivation is statistically significant ($F$ (3.402) = 61.015; $p < 0.001$). On the other hand, when analyzed in individual significance tests, it was found that only the persistence sub-dimension had a statistically significant and positive effect on internal motivation ($\beta = 0.336$; t (402) = 2.618; $p < 0.01$). The effects of independent variables on the dependent variable are listed as persistence ($\beta = 0.336$), proactivity ($\beta = 0.132$) and self-initiation ($\beta = 0.101$). Looking at the explanatory power of these three variables in the model, it was found that school administrators' personal initiative behaviors explained the internal motivation of teachers by 30.8% (Adjusted $R_2$ = 0.308). The findings suggested that only one predictor, $X_3$ (persistence), has the strong predictive power of Y (internal factors) with reference to both $\beta$ standardized coefficients. A one standard deviation increase in *persistence* would yield a 0.336 standard deviation increase in the predicted internal factors of teacher motivation with the other variables held constant.

**Table 6.** Examination of the Effect of Sub-Dimensions of Taking Personal Initiative on Internal, External, and Administrative Factor Dimensions of Teacher Motivation.

| Independent Variable | Internal Factors | | | External Factors | | | Administrative Factors | | |
|---|---|---|---|---|---|---|---|---|---|
| | β | t | p | β | t | p | β | t | p |
| Constant | 2.261 | | 0.000 | 2.389 | | 0.000 | 1.863 | | 0.000 |
| Self-Initiation | 0.101 | 0.651 | 0.516 | 0.230 | 1.470 | 0.142 | 0.484 | 3.669 | 0.000 |
| Proactivity | 0.132 | 0.856 | 0.392 | −0.176 | −1.137 | 0.256 | 0.354 | 2.705 | 0.007 |
| Persistence | 0.336 | 2.618 | 0.009 | 0.498 | 3.853 | 0.000 | −0.126 | −1.157 | 0.248 |
| R | | 0.559 | | | 0.552 | | | 0.711 | |
| R$_2$ | | 0.313 | | | 0.305 | | | 0.505 | |
| Adjusted R$_2$ | | 0.308 | | | 0.299 | | | 0.505 | |
| F | | 61.015 | | | 58.694 | | | 136.748 | |

$p < 0.05$.

As can be understood from Table 6, the regression model established to test the effect of the personal initiative dimensions (self-initiation, proactivity, persistence) of school administrators on the external factor dimension of teacher motivation is statistically significant ($F$ (3.402) = 58.694; $p < 0.001$). When analyzed in individual significance tests, it was found that only the persistence sub-dimension had a statistically significant and positive effect on external motivation ($\beta = 0.498$; t (402) = 3.853; $p < 0.01$). The effects of independent variables on dependent variables such as persistence ($\beta = 0.498$), self-initiation ($\beta = 0.230$) and proactivity ($\beta = -0.176$) were also assessed. Looking at the explanatory power of these three variables in the model, it was found that school administrators' personal initiative-taking behaviors explained the external motivation of teachers at a rate of 29.9% (Adjusted $R_2$ = 0.299). The results revealed that only one predictor, $X_3$ (persistence), has the strong predictive power of Y (external factors) with reference to both $\beta$ standardized coefficients. A one standard deviation increase in *persistence* would yield a 0.498 standard deviation increase in the predicted external factors of teacher motivation with the other variables held constant.

Finally, as seen in Table 6, the regression model established to test the effect of the dimensions of the personal initiative taking (self-initiation, proactivity, persistence) of the school administrators on the administrative factor dimension of teacher motivation is statistically significant ($F$ (3.402) = 136.748; $p < 0.001$). Considering the individual significance tests, the self-initiation ($\beta = 0.484$; t (402) = 3.669; $p < 0.01$) and proactivity ($\beta = 0.354$; t = 2.705; $p < 0.01$) sub-dimensions were found to have a statistically significant and positive effects on administrative motivation. The effects of the independent variables on dependent variables such as self-initiation ($\beta = 0.484$), proactivity ($\beta = 0.354$) and persistence ($\beta = -0.126$) were also assessed. Looking at the explanatory power of these three variables in the model, it was found that school administrators' personal initiative-taking

behavior explains teachers' administrative motivation by 50.1% (Adjusted $R_2 = 0.501$). The results suggested that predictors $X_1$ (self-initiation) and $X_2$ (proactivity) have the strongest predictive power of Y (administrative factors) with reference to both β standardized coefficients. A one standard deviation increase in *self-initiation* would yield a 0.484 standard deviation increase in the predicted administrative factors of teacher motivation with the other variables held constant. Similarly, a one standard deviation increase in *proactivity*, in turn, leads to a 0.354 standard deviation increase in the predicted administrative factors of teacher motivation with the other variables held constant.

In line with another sub-purpose of this study, we examined whether the subdimensions of the leadership styles of school administrators significantly predicted the internal, external and administrative factors of teacher motivation according to teachers' opinions. For this purpose, a regression model was created to examine the effect of the leadership style sub-dimensions of school administrators on the internal, external and administrative factors of teacher motivation. The analysis results for the model are presented in Table 7.

**Table 7.** Examination of the Effects of Leadership Style Sub-Dimensions on Internal, External, and Administrative Factor Dimensions of Teacher Motivation.

| Independent Variable | Internal Factors | | | External Factors | | | Administrative Factors | | |
|---|---|---|---|---|---|---|---|---|---|
| | β | t | *p* | β | t | *p* | β | t | *p* |
| Constant | 2.013 | | 0.000 | 2.499 | | 0.000 | 1.767 | | 0.000 |
| Democratic | 0.385 | 7.663 | 0.000 | 0.229 | 4.269 | 0.000 | 0.681 | 16.871 | 0.000 |
| Autocratic | 0.290 | 5.788 | 0.000 | 0.387 | 7.243 | 0.000 | 0.155 | 3.845 | 0.000 |
| Full Freedom | 0.183 | 4.494 | 0.000 | 0.027 | 0.627 | 0.531 | 0.050 | 1.539 | 0.125 |
| R | | 0.625 | | | 0.555 | | | 0.779 | |
| $R_2$ | | 0.391 | | | 0.308 | | | 0.607 | |
| Adjusted $R_2$ | | 0.386 | | | 0.303 | | | 0.604 | |
| F | | 85.927 | | | 59.683 | | | 207.089 | |

$p < 0.05$.

As can be understood from Table 7, the regression model established to test the effect of the leadership style dimensions of school administrators (democratic style, autocratic style, style allowing full freedom) on the internal factor dimensions of teacher motivation is statistically significant ($F$ (3.402) = 85.927; $p < 0.01$). Looking at individual significance tests, it was determined that all variables, democratic style (β = 0.385; t (402) = 7.663; $p < 0.01$), autocratic style (β = 0.290; t (402) = 5.788; $p < 0.01$) and full freedom style (β = 0.183; t (402) = 4.494; $p < 0.01$), significantly and positively affect internal motivation. Looking at the explanatory power of these three variables in the model, it was found that the leadership styles exhibited by school administrators explained the internal motivation of teachers by 38.6% (Adjusted $R_2 = 0.386$). The findings suggested that predictors $X_1$ (democratic), $X_2$ (autocratic), and $X_3$ (full freedom) have the strongest predictive power of Y (internal factors) with reference to both β standardized coefficients. A one standard deviation increase in *democratic style, autocratic style, and full freedom style* would yield a 0.385, 0.290, and 0.293 standard deviation increase in the predicted internal factors of teacher motivation, respectively, with the other variables held constant.

As seen in Table 7, the regression model established to test the effect of the leadership style dimensions of school administrators (democratic style, autocratic style, style with full freedom) on the external factor dimension of teacher motivation is statistically significant ($F$ (3.402) = 59.683; $p < 0.001$). When looking at individual significance tests, the variables democratic style (β = 0.229; t (402) = 4.269; $p < 0.01$) and autocratic style (β = 0.387; t (402) = 7.243; $p < 0.01$) were found to affect external motivation in a statistically significant and positive way. Looking at the explanatory power of these three variables in the model, it was determined that the leadership styles exhibited by school administrators explained the external motivation of teachers by 30.3% (Adjusted $R_2 = 0.303$). The results indicated that predictors $X_2$ (autocratic) and $X_1$ (democratic) have the strongest predictive power of Y (external factors) with reference to both β standardized coefficients. A one standard



deviation increase in *autocratic style* would yield a 0.387 standard deviation increase in the predicted external factors of teacher motivation with the other variables held constant. Similarly, a one standard deviation increase in *democratic style*, in turn, leads to a 0.229 standard deviation increase in the predicted external factors of teacher motivation with the other variables held constant.

As can be understood from Table 7, the regression model established to test the effect of the leadership style dimensions of school administrators (democratic style, autocratic style, style with full freedom) on the administrative factor dimension of teacher motivation is statistically significant ($F$ (3.402) = 207.089; $p < 0.001$). Looking at individual significance tests, the variables democratic style ($\beta$ = 0.681; t (402) = 16.871; $p < 0.01$) and autocratic style ($\beta$ = 0.155; t (402) = 3.845; $p < 0.01$) were found to have a statistically significant and positive effect on administrative motivation. Looking at the explanatory power of these three variables in the model, it was found that the leadership styles exhibited by school administrators explained the administrative motivation of teachers by 60.4% (Adjusted $R_2$ = 0.604). The results suggested that predictors $X_1$ (democratic) and $X_2$ (autocratic) have the strongest predictive power of Y (administrative factors) with reference to both $\beta$ standardized coefficients. A one standard deviation increase in *democratic style* would yield a 0.681 standard deviation increase in the predicted administrative factor dimension of teacher motivation with the other variables held constant. Similarly, a one standard deviation increase in *autocratic style*, in turn, leads to a 0.155 standard deviation increase in the predicted external factors of teacher motivation with the other variables held constant.

## 4. Discussion, Conclusions, and Suggestions

According to the results obtained from the research findings, it was observed that the motivation levels of the teachers were high. In terms of the sub-dimensions of motivation, it was determined that teachers' internal motivation levels were "high", and their external and administrative motivation levels were "quite high". According to this, the individual motivation levels of teachers are high even if the administrators have no positive contributions.

In this study, it was determined that there is a significant and positive relationship between school administrators' personal initiative-taking behavior and teachers' motivation. Therefore, it can be said that the motivation level of teachers increases as school administrators take personal initiative regarding their duties. Considering similar studies in the literature, other findings supporting this result have been achieved. Kılıç [18], in his research on teachers working in primary, secondary and high schools in Konya province, concluded that there is a significant and positive relationship between the dimensions of self-initiation, proactivity and the persistence of school administrators' initiative-taking behaviors and the intrinsic, extrinsic and administrative factors of teacher motivation. Similarly, Kurt [45] examined the effect of the leadership characteristics of school administrators in secondary education institutions on teacher motivation, determining that there is a significant and positive relationship between the perception of the leadership behaviors of administrators on teachers and teachers' motivation levels and stating that teachers' level of motivation is directly related to the leadership skills of their administrators. On the other hand, Jaramillo et al. [46] found that there was a significant relationship between organizational administrators' personal initiative levels and employees' internal motivation, and found that personal initiative assumes a mediating role between employees' internal motivation and job performance. In the studies of Tornau and Frese [47], it was found that there is a positive link between personal initiative and business success. Stroppa and Speiss [48] stated in their study that employees who take personal initiative with the support of their managers are more successful in their jobs and exhibit active performance, meaning that their job success is higher.

Another result obtained from the findings of this study is that there is a significant relationship between school administrators' leadership styles and teachers' motivation. According to teachers' views, there is a positive and significant relationship between the

democratic and autocratic style sub-dimensions of school administrators and the internal, external and administrative factors of teacher motivation. On the other hand, no significant relationship was found between internal and external factors of teacher motivation in the full freedom sub-dimension, and a low level/negative relationship was found in the dimension of administrative factors. It is expected that democratic and autocratic leadership styles are effective in teacher motivation. Quite different results have been obtained in various studies in the literature. For example, in Adeyemi's [49] study, it is stated that the job performance of teachers in schools where the autocratic leadership style is used is higher than that in schools where democratic and full freedom leadership styles are used. Adjei and Amofa [50] noted that teachers who work with school administrators who demonstrate democratic management have a high level of motivation, especially in terms of their participation in the decision-making process. In another study, Franklin [51] emphasizes that leadership styles affect teachers' internal and external motivations differently and provide practical implications for school administrators. Kırıştı [52] stated, in his research, that there is a positive and significant relationship between the behaviors of school administrators and teachers' motivation levels, and this relationship contributes significantly to an increase in teachers' motivation levels by increasing their satisfaction levels. Ugar [53] found that there is a positive and low-level relationship between the leadership practices of school principals and teachers' motivation. Ada et al. [54] investigated the internal and external factors that motivate classroom teachers and affect teachers' motivation negatively. In the study, it was found that external motivation tools were most effective in teacher motivation. It has been determined that teachers want to work with powerful and trustworthy school administrators in order to be successful in their school duties. Arslan [55] concluded, in his study, that there is a moderately significant positive correlation between school administrators' democratic attitudes and behaviors and teacher motivation. Furthermore, the fact that a leadership style with full freedom harms teacher motivation is quite consistent with the findings in the literature. In the study by Gopal and Chowdhury [56], it was determined that leadership style with full freedom harms the motivation of employees. In a similar study by Alasad [57], a negative relationship was found between school administrators who allow full freedom and the internal and external motivations of teachers.

As can be understood from the research findings, while democratic and autocratic leadership behaviors affect teacher motivation positively, albeit at a low level, leadership behaviors that allow full freedom affect teacher motivation negatively. In the study of Kılıç and Yılmaz [42], the effect of school principals' leadership styles on teachers' motivation was investigated and it was emphasized that teachers prefer to work with administrators who have a democratic attitude and that they want their administrators' behavior and attitudes to be democratically oriented. Franklin [51] examined the preferences of U.S. teachers regarding administrators' leadership styles. In the study, it was emphasized that teachers mostly prefer a leadership style that allows full freedom, and that leadership styles of administrators should motivate teachers to reach their goals, thus contributing to social change. In the study of Arslan [55], it was determined that there is a significant relationship between the democratic attitudes and behaviors of school administrators and teacher motivation. As a result, it is understood that the leadership styles exhibited by school administrators affect teachers' motivation directly or indirectly.

Considering that administrators' personal initiative-taking behaviors affect the internal, external and administrative factors, which are the dimensions of teacher motivation, it was seen that the persistence dimension was a significant predictor of internal and external factors. Persistence refers to acting with a determined and patient approach while accomplishing a goal or task. Ponton [58] defines persistence as the behavior of continuing with an action despite obstacles. For administrators, persistence means not giving up when dealing with problems and opportunities and constitutes an important component of personal initiative. As can be understood from the results of the research, when administrators who take personal initiative in a management task behave "persistently", they also affect

teachers' motivation positively. In addition to persistence, an important component of effective leadership is the necessity of persistent behavior in a logical framework [25]. Persisting with an incorrect strategy can hurt the organization [59]. Moreover, such behaviors, when exhibited by managers, can be perceived as stubborn by the employee, rather than persistent. The persistence debate in the personal initiative literature shows that this point has been neglected.

In this study, it was determined that the sub-dimensions of the self-initiative and proactivity of school administrators' behaviors were significant predictors of the administrative factor dimensions of teacher motivation. Administrative factors, as motivational tools, include the leadership attitudes and behaviors that school administrators exhibit towards teachers. For example, the fact that administrators are sensitive and helpful, support and appreciate teachers' work, encourage them to do new studies, provide a peaceful working environment, and gain their trust can be considered as important administrative factors for teachers. In their research, Joo and Lim [60] reported that the higher the proactive personalities of employees, the higher their tendency to feel motivated. Similarly, in the study conducted by Fay and Frese [20], it was stated that one of the most important factors for achieving success in personal initiative-related behaviors in the workplace is motivation. It was emphasized that taking personal initiative is closely related to the performance and motivation of those who work in various settings, and also that personal initiative controls motivated behavior.

Although the structure of the current education system in Turkey is not suitable for taking personal initiative, the behavior of taking personal initiative is a process that is directly related to teacher motivation, affecting it in a positive way. Therefore, the initiative-taking behavior of school administrators will directly affect the quality and characteristics of the education provided by teachers. Administrators taking the initiative can transform the hierarchical and bureaucratic functioning of schools into strategic situations that benefit schools and enable action to be taken. From this point of view, it is clear that taking initiative is not only an individual behavior, but also an organizational and administrative behavior [61].

In this study, in the regression model established to examine the effect of the leadership styles of school administrators on the internal factor dimensions of teacher motivation, democratic, autocratic, and full-freedom leadership styles predict internal motivation. On the other hand, the democratic and autocratic style variables exhibited by the administrators predict both external factors and administrative factors. It is understood that the style variable that gives full freedom does not have a significant effect on administrative factors. According to Mruma [62], although both internal and external motivations are accepted as important motivating factors by teachers, internal factors are the most effective motivation factors for teachers. Considering the general results of the study, it is understood that teachers' internal motivation levels are high, and that the leadership styles of administrators are important predictors of their motivation.

As with all other studies, this study had some limitations. The research was planned as a mixed-methods approach, including quantitative and qualitative methods. However, in the process of data collection in Turkey, due to COVID-19 restrictions, we were unable to perform interviews in order to collect the qualitative data. Another limitation of this study is the lack of research data that reflects Turkey in general. This research is limited only to data obtained from high schools in Manisa province. A survey should be conducted on a sample representing all geographical segments in Turkey, as we believe that future research should include more generalizable results. Furthermore, we suggest that the data obtained through a mixed research methods approach will add new dimensions to our research, which may be useful to other researchers.

In the final analysis, school administrators should be more sensitive and more aware of the needs of teachers, because administrators must keep their enthusiasm and interest alive in order to motivate teachers [63]. School administrators that respond to the personal needs of teachers, encourage team building among employees, motivate teachers as role

models and pay personal attention to teachers evidently increase teachers' job performance and motivation levels [64]. Teachers' motivation factors are prominently involved in the sustainable development of educational institutions [65]. Managers must provide effective leadership that motivates teachers to energize and educate students, as well as stimulating desire and enthusiasm in teachers [66]. The duty of school administrators is not only to respond to the demands of the legislation that has emerged as a result of today's standards, but also to take responsibility for their teachers' motivational needs.

This study provides evidence that school administrators' personal initiative behavior is directly related to teacher motivation and that this positively affects teacher mobility, thereby affecting the quality of their teaching. The structure of the current education system in Turkey does not promote personal initiative, despite the fact that personal initiative-taking behavior is directly related to teacher motivation in a positive manner. In light of the results obtained, it can be assumed that personal initiative is a behavior that has organizational and administrative effects beyond simply being an individual behavior. An understanding of education that is focused on sustainable development should not be seen as an awareness concept, but as a concept that has social, cultural and economic effects.

**Author Contributions:** Conceptualization, G.D.; Z.A.; methodology, G.D.; F.A. and S.Y.; software, S.Y.; Ü.K.; validation, G.D.; S.Y. and Ü.K.; formal analysis, Ü.K. and S.Y.; investigation, G.D.; F.A. and Z.A.; resources, G.D. and Z.A.; data curation, Ü.K. and S.Y.; writing—original draft preparation, S.Y. and Ü.K.; writing—review and editing, S.Y.; Ü.K., G.D., Z.A. and F.A.; visualization, S.Y.; supervision, G.D.; project administration, G.D.; funding acquisition, S.Y. All authors have read and agreed to the published version of the manuscript.

**Funding:** This research received no external funding.

**Informed Consent Statement:** Informed consent was obtained from all subjects involved in the study.

**Data Availability Statement:** MDPI Research Data Policies rules have been checked. The study is prepared in accordance with these rules.

**Conflicts of Interest:** The authors declare no conflict of interest.

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
