# Peer review of "The Effect of Leadership Styles and Initiative Behaviors of School Principals on Teacher Motivation"

_sustainability, doi:10.3390/su13052711_

Round 1

Reviewer 1 Report

Here are my comments:

1 There is no need to to repeat what you demonstrated in Table 2, so you can either keep table 2 or lines 305-323. I suggest to keep the table instead of the description.

2 I saw you reported alpha by sub-dimension, do you have an alpha for each dimension and an alpha for the instrument as a whole?

3 Any reference for supporting ", their personal initiative taking behaviors and the motivation levels of the teachers, the ranges are 1.00-1.79 "quite low", 1.80-2.59 "low", 2.60-3.39 "medium", 3.40-4.19 "high", the range of 4.20-5.00 was evaluated as "quite high".

4 3.3 findings related to regression analysis is very confusing. Take the results in table 6 as an example, the dependent variable is internal teacher motivation. what is independent variable? the aggregated score of personal initiative taking or its sub dimensions (self-initiation, proactivity, and persistence)?

5 The author might want to check structure equation modelling for analyzing this dataset. Should internal, external, and administrative factors be correlated since they also belong to teacher motivation? If so, why authors want to analyze them separately?

Author Response

Response to Reviewer 1 Comments:

First of all, thank you for reviewing my research article and for your comments. I share my answers and arrangements for your comments.

Point 1: There is no need to to repeat what you demonstrated in Table 2, so you can either keep table 2 or lines 305-323. I suggest to keep the table instead of the description.

Response 1: The explanations section of Table 2 has been removed in line with the reviewer's comments. [in line: 274]

Point 2: I saw you reported alpha by sub-dimension, do you have an alpha for each dimension and an alpha for the instrument as a whole?

Response 2:  Cronbach Alpha values of all scales used in the study were added. [in line: 289;352;379]

Point 3: Any reference for supporting ", their personal initiative taking behaviors and the motivation levels of the teachers, the ranges are 1.00-1.79 "quite low", 1.80-2.59 "low", 2.60-3.39 "medium", 3.40-4.19 "high", the range of 4.20-5.00 was evaluated as "quite high".

Response 3: Reference to value ranges has been added. [in line: 422]

Point 4: “4 3.3 findings related to regression analysis is very confusing. Take the results in table 6 as an example, the dependent variable is internal teacher motivation. what is independent variable? the aggregated score of personal initiative taking or its sub dimensions (self-initiation, proactivity, and persistence)?”

Response 4: This section has been completely revised according to suggestion. [in line: 474 -562]

Point 5: 5 The author might want to check structure equation modelling for analyzing this dataset. Should internal, external, and administrative factors be correlated since they also belong to teacher motivation? If so, why authors want to analyze them separately?

Response 5: Thank you for your suggestion to design this data set and research in a structural equation model. I had concerns that it would cause too much change in theory and practice, especially in the aims of the article. however, I would like to state that you are right in your suggestion.

Point 6: “English language and style are fine/minor spell check required”

Response 6: According to the comments of the reviewers, necessary language arrangements have been made in my research article

If you wish, I would be happy to answer your questions about the answers I gave to your comments. thanks

Reviewer 2 Report

Thank you very much for allowing me to review this article on The Effect of Leadership Styles and Initiative Behaviors of School Principals on Teacher Motivation.
Introduction. This section should be restructured so that it is not so long. It should be more compact and focused on the three domains it addresses. There are paragraphs in which there is no significant contribution, just personal comments from the authors.
On line 271 appears the sentence "interventionary studies involving animals or humans, and other studies that require ethical approval, must list the authority that provided approval and the corresponding ethical approval code." Not much sense. Check this fact.

The paragraph below table 2 is not necessary as stated. It makes it difficult to read. If you want, you can mention the most significant data, but without having to mention the percentages that are already included in table 2.
Section 4 should be called Discussion, conclusions and suggestions.
This section should be revised and not include the three dimensions, discussion, conclusions and suggestions. I think it should include the limitations of the study, as well as the theoretical and practical implications of the study.

Author Response

Response to Reviewer 2 Comments:

First of all, thank you for reviewing my research article and for your comments. I share my answers and arrangements for your comments.

Point 1: “Introduction. This section should be restructured so that it is not so long. It should be more compact and focused on the three domains it addresses. There are paragraphs in which there is no significant contribution, just personal comments from the authors.”

Response 1: The introduction has been revised to focus on the area it addresses. Paragraphs reflecting the personal comments of the authors were removed and restructured. [in line: 77-212]

Point 2: “On line 271 appears the sentence "interventionary studies involving animals or humans, and other studies that require ethical approval, must list the authority that provided approval and the corresponding ethical approval code." Not much sense. Check this fact.”

Response 2: The part about ethical approval has been removed. [in line: 238]

Point 3: “The paragraph below table 2 is not necessary as stated. It makes it difficult to read. If you want, you can mention the most significant data, but without having to mention the percentages that are already included in table 2.”

Response 3: The explanations section of Table 2 has been removed in line with the reviewer's comments. [in line: 274]

Point 4: “Section 4 should be called Discussion, conclusions and suggestions. This section should be revised and not include the three dimensions, discussion, conclusions and suggestions. I think it should include the limitations of the study, as well as the theoretical and practical implications of the study.”

Response 4: Chapter 4 has been renamed "discussion, conclusions and recommendations" according to the reviewer's comments. In addition, a paragraph on research limitations has been added to this section. [in line: (Chapter,563); (limitations, 693-701)]

Point 5: “English language and style are fine/minor spell check required”

Response 5: According to the comments of the reviewers, necessary language arrangements have been made in my research article

If you wish, I would be happy to answer your questions about the answers I gave to your comments. thanks

Reviewer 3 Report

The aim of this study is to investigate the effect of school administrators' personal initiative behaviors and their leadership styles on teacher motivation.  Pearson  correlation technique and multiple regression analysis technique were used to determine whether there was a significant relationship between dependent and independent variables during the data analysis. According to the results authors  found that the general motivation levels of the teachers were high. In the analyzes conducted in terms of the relationships between school administrators' personal initiative taking behaviors and teachers' motivation, it was found that there was a significant and positive relationship.

  This work presents a novel topic and brings interesting conclusions to the literature. The work is well structured and regarding the proposed methodology and the results, one of the limitations is that the authors have not delved into the possible mediating or moderating variables that could affect the motivation of teachers. He also suggested that they expand the description of their sample The conclusions should emphasize the practical implications of your study highlighting the contribution to the literature

Author Response

Response to Reviewer 3 Comments:

First of all, thank you for reviewing my research article and for your comments. I share my answers and arrangements for your comments.

Point 1: “The aim of this study is to investigate the effect of school administrators' personal initiative behaviors and their leadership styles on teacher motivation.  Pearson correlation technique and multiple regression analysis technique were used to determine whether there was a significant relationship between dependent and independent variables during the data analysis. According to the results authors found that the general motivation levels of the teachers were high. In the analyzes conducted in terms of the relationships between school administrators' personal initiative taking behaviors and teachers' motivation, it was found that there was a significant and positive relationship.

  This work presents a novel topic and brings interesting conclusions to the literature. The work is well structured and regarding the proposed methodology and the results, one of the limitations is that the authors have not delved into the possible mediating or moderating variables that could affect the motivation of teachers. He also suggested that they expand the description of their sample The conclusions should emphasize the practical implications of your study highlighting the contribution to the literature”

Response 1: Due to word limitation, more focus has been implemented to the results in the summary section. According to the reviewer's comments, the summary part has been revised and restructured. The brief re-added about the sampling has been added to the summary part. Some additions were made to the summary and findings section about dependent and independent variables that affect teachers' motivation. Practical results emphasizing the contribution of the research to the literature are added in line with your comments. [in line: (abstract,16), (conclusions,714-719)]

Point 2: “English language and style are fine/minor spell check required”

Response 2: According to the comments of the reviewers, necessary language arrangements have been made in my research article.

If you wish, I would be happy to answer your questions about the answers I gave to your comments. thanks

Round 2

Reviewer 1 Report

Here is my comments:

#1 p should be italic. Line 489 and the rest.

#2 R2 should be R2, please edit them accordingly.

#3 The author might want to explain what "democratic style (β=.385), autocratic style (β=.290), and full freedom style (β=-.183)" means, in your research context, to help the audience to understand. 

#4 The author might want to relate your study/topic more to sustainability to fit the scope of this journal, e.g., in the literature review, discussion, and conclusion.

Author Response

This section has been completely revised according to suggestion. [in line: 425 -431]

This section has been completely revised according to suggestion. [in line: 686 -694]

Necessary language arrangements have been made in my research article.

If you wish, I would be happy to answer your questions about the answers I gave to your comments. thanks.

Reviewer 2 Report

Thank you very much for responding to the comments and suggestions. I think the article now has greater strength and consistency. 

Author Response

Necessary language arrangements have been made in my research article.

If you wish, I would be happy to answer your questions about the answers I gave to your comments. thanks.